# Knowledge of Self-Isolation Rules in the UK for Those Who Have Symptoms of COVID-19: A Repeated Cross-Sectional Survey Study

**DOI:** 10.3390/ijerph20031952

**Published:** 2023-01-20

**Authors:** Louise E. Smith, Robert West, Henry W. W. Potts, Richard Amlȏt, Nicola T. Fear, G. James Rubin, Susan Michie

**Affiliations:** 1Institute of Psychiatry, Psychology and Neuroscience, King’s College London, London SE5 9RJ, UK; 2NIHR Health Protection Research Unit in Emergency Preparedness and Response, London SE5 9RJ, UK; 3Department of Behavioural Science and Health, University College London, London WC1E 6BT, UK; 4Institute of Health Informatics, University College London, London NW1 2DA, UK; 5Behavioural Science and Insights Unit, UK Health Security Agency, London SW1P 3HX, UK; 6Academic Department of Military Mental Health, King’s Centre for Military Health Research, London SE5 9RJ, UK; 7Centre for Behaviour Change, University College London, London WC1E 7HB, UK

**Keywords:** COVID-19, knowledge, understanding, self-isolation, regulations

## Abstract

Objectives: To investigate knowledge of self-isolation rules and factors associated with knowledge. Methods: Repeated cross-sectional online surveys (n ≈ 2000 UK adults) between 9 November 2020 and 16 February 2022 (78,573 responses from 51,881 participants). We computed a composite measure of knowledge of self-isolation rules and investigated associations between knowledge and survey wave, socio-demographic characteristics (age, gender, UK nation, index of multiple deprivation), trust in government, and participants’ belief that they had received enough information about self-isolation. Results: In total, 87.9% (95% CI 87.7% to 88.1%, n = 67,288/76,562) of participants knew that if they had symptoms of COVID-19 they should ‘self-isolate’. However, only 62.8% (n = 48,058/76,562, 95% CI 62.4% to 63.1%) knew the main rules regarding what that meant. Younger people had less knowledge than older people, and men had less knowledge than women. Knowledge was lower in people living in England versus in Scotland, Wales, and Northern Ireland. The pattern of association between knowledge and trust in government was unclear. Knowledge was lower in people living in a more deprived area and those who did not believe they had enough information about self-isolation. Knowledge was lower in December 2020 to January 2021, compared with before and after this period. Conclusions: Approximately 63% of UK adults between November 2020 and February 2022 appeared to know the main rules regarding self-isolation if symptomatic with COVID-19. Knowledge was lower in younger than older people, men than women, those living in England compared with Scotland, Wales or Northern Ireland, and those living in more deprived areas.

## 1. Introduction

In the UK, the self-isolation of people with symptoms of COVID-19 or a positive test was one of the main strategies used to prevent the spread of infection. Since 12 March 2020, anyone in the UK with the main symptoms of COVID-19 (as described by the National Health Service [NHS]: high temperature and new, continuous cough) was required to self-isolate for at least 7 days (Figure 1) [1]. They should also not have had visitors to their home. On 18 May 2020, loss or change to sense of smell or taste was added to the key symptoms [2], and people with symptoms were required to undergo a polymerase chain reaction (PCR) test to check if they had COVID-19, starting from 18 May 2020 [3,4,5]. From this date, people had to stay at home until they received their test result and could only leave their home to have a test [3]. The isolation period was extended to 10 days on 30 July 2020 [6]. Self-isolation became a legal requirement for those with COVID-19 symptoms on 28 September 2020 [7]. Changes to isolation rules were made on 16 August 2021 (removing the need for vaccinated contacts of cases to isolate) [8], 22 December 2021 (shortening the isolation period from 10 days to 7 days provided a negative LFD test on day 6 and 7 of symptoms) [9], 11 January 2022 (removing the need for people who tested positive using a lateral flow device [LFD] to have a confirmatory PCR test) [10], and 17 January 2022 (shortening the isolation period to 5 days following a negative test on day 5 and 6 of symptoms) [11], and the legal requirement to self-isolate was lifted on 24 February 2022 [12].

Surveys suggest that over 50% of people with COVID-19 symptoms in the UK failed to adhere fully to these rules [13,14]. This will have contributed to high transmission rates. Lack of knowledge of the rules and procedures for self-isolation were associated with lack of adherence to self-isolation rules [13,14,15]. Research has shown that knowledge of rules for self-isolation was low among the general public in the UK [14], but higher among the subset of symptomatic people who were known to NHS Test and Trace [16]. Survey data can sometimes be sensitive to small changes in item wording. However, if knowledge is low, then understanding factors influencing knowledge can inform policies and interventions for improving knowledge of and adherence to self-isolation rules.

The devolved nations (Scotland, Wales and Northern Ireland) have differed in some of their rules to reduce COVID-19 spread [17,18]. Initially, self-isolation rules were the same across the UK, but changes to rules happened at different times across the four nations (e.g., Senedd Research 2022 [19], Scottish Parliament 2022 [20], Institute for Government 2022 [21]). If knowledge of self-isolation rules varies substantially across nations, this could provide a basis for learning from those that perform better. Research throughout the pandemic has shown that adherence to rules is lower in certain sociodemographic groups, such as men and younger people [13,22]. Investigating whether knowledge of rules is also lower in these groups could identify target groups for interventions and inform communications aimed at increasing knowledge. Lack of trust in the government has also been associated with non-adherence [22,23]. This may suggest another way in which knowledge could be improved, for example by increasing trust in government or attributing messaging to sources other than government such as the NHS.

The aims of this study were to investigate the UK public’s knowledge of self-isolation rules if one has COVID-19 symptoms, and whether knowledge varies by nation, socio-demographic characteristics (gender, age, and deprivation), trust in government, participants’ beliefs that they have received enough information about self-isolation, and changes over time. We also investigated whether a change in question wording affected responses.

## 2. Materials and Methods

### 2.1. Design

A series of cross-sectional, UK-wide, nationally representative surveys (weekly or fortnightly) were conducted by BMG Research and then Savanta on behalf of the Department of Health and Social Care over the course of the COVID-19 pandemic. We analyzed these data as part of the CORSAIR study (the COVID-19 Rapid Survey of Adherence to Interventions and Responses study) [13]. For the current paper, we used data from wave 32 to wave 68 (9 November 2020 to 16 February 2022). We selected these waves as there were no major changes to questions about knowledge of self-isolation rules during this period.

### 2.2. Participants

Participants (n ≈ 2000 per wave) were recruited from two specialist research panel providers, Respondi (n = 50,000) and Savanta (n = 31,500). They were eligible for inclusion if they were aged 16 years or over and lived in the UK. Quotas were applied based on age and gender (combined). After completing the survey, participants were then unable to participate in the subsequent three waves. Participants were assigned unique identifier codes to determine repeat participation. Participants were reimbursed in points that could be redeemed in cash, gift vouchers or charitable donations (up to 70p per survey wave).

### 2.3. Study Materials

#### 2.3.1. Outcome Measures

To measure knowledge about government rules and procedures for those with symptoms of COVID-19, we asked participants a series of true/false questions regarding statements about actions that people should or could carry out “if you have symptoms of coronavirus”. Statements included self-isolating, being able to go out—to the shops for essentials, to the shops for non-essentials, for exercise, for work, and to meet people from another household—as well as being able to go out if you were wearing a face covering or if symptoms were only mild (see Appendix A for full items).

We computed a knowledge score by combining responses to four items that would involve the contact of a symptomatic person with people from other households in an indoor space (going out to the shops for essential items, going out to the shops for non-essential items, to meet up with friends or family indoors, and having someone over to one’s home). Participants were coded as being “correct” for this item if they identified that all four answers were false.

The wording of the knowledge question focused on what the respondent should or could do (“if you have symptoms of coronavirus, you…”). This may have led respondents to personalize their responses, for example, correctly asserting that they need not self-isolate if they have had a PCR test which has come back negative. To test this, in waves 47 and 48, half the sample, selected at random, received an alternative version of the wording (“If someone develops symptoms of coronavirus, they…”) and half received the original wording.

#### 2.3.2. Explanatory Factors

We asked participants for their age, gender, what country they lived in, and full postcode. From their postcode, we derived index of multiple deprivation (in quartiles) [24].

Participants were asked to what extent they agreed that information from the government about coronavirus could be trusted, and that they had enough information from the government and other public authorities on self-isolation. Both questions were answered on a five-point scale from “strongly agree” to “strongly disagree”.

### 2.4. Ethics

This work was undertaken as part of service evaluation of the marketing and communications conducted by the Department of Health and Social Care, and, following advice from King’s College London Research Ethics Subcommittee, was exempt from requiring ethical approval.

### 2.5. Power

A sample size of 2000 allows a 95% confidence interval of plus or minus 2% for the prevalence estimate for a survey item with a prevalence of around 50%. We then pooled data from multiple survey waves, providing greater power for analyses of relationships between variables.

### 2.6. Analysis

Prevalence of knowledge of rules if symptomatic was assessed using percentages and 95% confidence intervals.

We conducted a generalized estimating equation (GEE) to investigate associations between socio-demographic factors (nation, gender, age, and index of multiple deprivation), psychological factors (trust in the government, and perceived sufficiency of information about self-isolation), survey wave and key knowledge scores, adjusting for all variables (*p* < 0.05 2-tailed). The GEE adjusts for individuals completing the survey on more than one occasion. There were some participants for whom unique participant identifiers were not assigned (1.4%, n = 1087/78,573). Participants who were not assigned a unique participant number were not included in the GEE analysis.

To investigate whether changing the phrasing of the question affected responses, we conducted χ^2^ tests, comparing responses to the amended version (“If someone develops symptoms…”) of the knowledge items to the original wording (“If you have symptoms…”; waves 47 and 48 only).

Data were weighted based on age, gender and Government Office Region to reflect targets based on data from the Office for National Statistics [25]. The GEE was conducted on unweighted data.

## 3. Results

There were 78,573 responses included in the weighted sample, from at least 51,881 participants. In total, 50.9% of respondents were female (n = 40,006/78,573; 48.7% male, n = 38,274; 0.3% preferred to self-describe, n = 222; and 0.1% preferred not to say, n = 72). The mean age was 47.7 years (standard deviation 18.6 years, range 16 to 93 years). Additionally, 82.7% (n = 64,955) identified as white British (6.2% white other, n = 4853; 2.5% mixed, n = 1992; 5.1% Asian or Asian British, n = 4001; 2.4% Black or Black British, n = 1909; 0.5% Arab or other, n = 399, 0.6% preferred not to say, n = 463).

While 87.9% (95% CI 87.7% to 88.1%, n = 67,288/76,562 of responses identified that if participants had symptoms of COVID-19, they should self-isolate, knowledge for individual statements varied (Table 1) (in waves 47 and 48, half of the sample received an alternative version of the wording and are therefore not included in these analyses). Of all participants, 62.8% (n = 48,058/76,562, 95% CI 62.4% to 63.1%) scored correctly on our derived knowledge measure. The most incorrectly answered statements were that if you had COVID-19 symptoms, you should take a lateral flow test (14.9% correct) and that you could be eligible for self-isolation payments (59.6% correct). For out-of-home behavior, the most incorrectly answered statements were that if you had symptoms of COVID-19, you could go out for a walk or some other exercise (62.0% answered correctly) and spend time outdoors for recreational purposes (68.6% answered correctly).

Correct knowledge on the composite knowledge measure was associated with being older, female, living in Scotland, Wales, and Northern Ireland (compared with England), and disagreeing that information from the Government about COVID-19 could be trusted (Table 2). Incorrect knowledge was associated with living in a more deprived area and disagreeing that you had enough information from the government and other public authorities about self-isolation. Analyses investigating survey wave indicated that compared to early November 2020, knowledge was lower in December 2020 to January 2021 and higher between September 2021 and February 2022.

### Comparison of Item Phrasing

Changing the phrasing of the question from “if you have symptoms of coronavirus, you…” to “if someone develops symptoms of coronavirus, they…” had only a small effect on participant responses (Table 3). The percentage of respondents correctly responding to statements concerning going to the shops for groceries/pharmacy, going out for exercise, caring for a vulnerable person, going out if wearing a face covering, and going out to meet people from another household outdoors was slightly higher when the question was phrased to refer to another person than when phrased to refer to themselves.

## 4. Discussion

Only 63% of respondents correctly identified four key self-isolation rules, although 88% knew in some sense that they should self-isolate if they had symptoms of COVID-19. These rules were mandated by law between September 2020 and February 2022, having previously been recommendations [1,26]. The most common non-adherent behaviors endorsed as something that people should or could do when symptomatic were going out for exercise, to spend time outdoors for recreational purposes and for essential shopping. Apart from spending time outdoors for recreation, these were actions that were allowed for the general population under lockdown measures [27,28,29], but not under self-isolation rules. Other little-understood rules included not going out to help or care for vulnerable people, and not going out to work where you could not work from home. This is important given the potential spread to those most clinically vulnerable to COVID-19 and to the large number of contacts made in the workplace [30,31].

We found lower rates of understanding of self-isolation rules than another study, which found that 80% of people who had tested positive for SARS-CoV-2 and were in the NHS Test and Trace system fully understood self-isolation requirements [16]. However, many people with COVID-19 symptoms did not request a test, or completed only lateral flow tests, and therefore did not enter the NHS Test and Trace system [13,32]. Ensuring that everyone understands the basic principles of self-isolation means that even those who did not enter the test, trace, and isolate system would have been well-placed to adhere to self-isolation rules, reducing onward transmission.

Understanding of self-isolation rules was lower in December 2020 to January 2021 compared to early November 2020. This may have been due to confusion over changing rules over the Christmas period. Understanding also appeared to increase from mid-September 2021. However, this increase aligns with the change in market research company. An unknown, subtle change in the question formatting during the change may have been responsible. Changes to self-isolation rules were made in August 2021, with a series of changes made in December 2021 to January 2022, followed by the removal of the legal obligation to self-isolate. These changes did not appear to affect understanding about self-isolation rules.

Groups with lower knowledge of self-isolation rules were younger people, men, and those living in more deprived areas. These groups have consistently been found to have lower adherence to rules throughout the pandemic [13,14,22]. Lack of knowledge has been found to contribute to lower adherence [13,14,15]. Congruent with this is our finding that knowledge of self-isolation rules was lower among people who disagreed that they had enough information about self-isolation. Knowledge was also worse among people living in England compared to other UK nations. We are not sure why this is the case. In future, communication strategies to increase knowledge of self-isolation rules should be designed to reach groups with lower knowledge, ensuring that the correct channels, media and messengers are being employed to promote engagement.

The UK Government introduced GBP 500 support payments for people who meet specific criteria on 28 September 2020 to increase adherence in areas of higher deprivation [26]. It was striking that the percentage of people responding ‘don’t know’ to our item asking about knowledge of this payment far exceeded ‘don’t know’ responses for any other item, suggesting a high degree of uncertainty in the population. Support payments can only encourage people to engage with self-isolation systems if people are aware that they exist and that they might be eligible for them. Government estimates suggest that, in England, under 4 million people were eligible for this support payment [26] (total England population 56.5 million [33]). Therefore, while knowledge was low, only a few people in the study may be eligible for this payment. Clearer communication about this should be a priority.

The pattern between knowledge of self-isolation and believing that information from the government about COVID-19 can be trusted was not clear. Although analyses identified a significant difference, in practice, our large sample size allowed us to detect small differences as significant. In this case, rates of knowledge were similar for those who strongly agreed (56%) and strongly disagreed (60%) that information could be trusted.

Changing the phrasing of the wording of a question relating to self-isolation from being personally relevant to relating to someone else had a small impact on responses. While participants were statistically significantly better at correctly responding to statements when the question was phrased to relate to “someone” rather than themselves, in practice, they only differed by up to three percentage points.

One strength of this study is that it used a large sample, weighted to represent the UK population. This means that we were able to detect very small differences within the data. Limitations include the use of an online survey. We cannot be sure that online panel respondents are representative of the UK population in terms of knowledge and beliefs. However, associations within the sample are likely to mirror those in the general population [34]. It is also relevant that, because the sample received payments to take part in a wide range of market research surveys, they were unlikely to have any specific interest in COVID-19. This will have reduced the likelihood of bias due to topic interest among participants. Surveys were cross sectional, and therefore we cannot imply a direction of causation. Rates of understanding of self-isolation increased when the survey was conducted by a different market research company, despite the sampling strategy remaining the same. As survey wave was included in adjusted regressions, we controlled for this effect, and it therefore does not affect our interpretation of other results. Due to space limitations in the survey questionnaire, we were unable to investigate the credibility of individual health messages.

## 5. Conclusions

Full adherence to self-isolation in people who have symptoms of COVID-19 is low. This is in part probably due to incorrect knowledge about self-isolation rules. While people know they should self-isolate if symptomatic, knowledge of individual rules is variable. Incorrect knowledge of self-isolation rules, particularly those in which a symptomatic individual would come into contact with people from other households indoors, could contribute to virus transmission as people continue to carry out everyday activities. Knowledge was lower in younger people, men, people living in England, and people living in more deprived areas. Increasing knowledge about self-isolation rules and procedures may increase adherence to self-isolation and decrease virus transmission.

## Figures and Tables

**Figure 1 ijerph-20-01952-f001:**
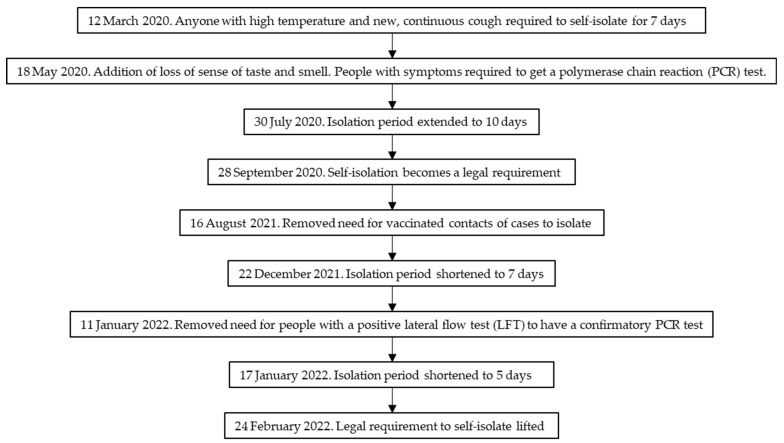
Flowchart of changes to self-isolation rules.

**Table 1 ijerph-20-01952-t001:** Endorsement of individual statements for knowledge of self-isolation (correct answer shown in bold).

The Government Has Issued Advice on How People Should Help Prevent the Spread of Coronavirus if They Have Symptoms. If You Have Symptoms of Coronavirus, You: [Total n = 76,562]	True, % (n)	False, % (n)	Don’t Know, % (n)
Should self-isolate	**87.9 (67,288)**	7.5 (5773)	4.6 (3501)
Could be eligible for self-isolation payments (if you are on benefits and/or a lower income and cannot work from home)	**59.6 (45,642)**	11.9 (9120)	28.5 (21,800)
Can go out if your symptoms are mild	9.0 (6899)	**84.6 (64,780)**	6.4 (4883)
Can go to the shops for groceries/pharmacy *	23.0 (17,593)	**71.6 (54,808)**	5.4 (4160)
Can go to the shops for things other than groceries/pharmacy *	16.2 (12,370)	**78.7 (60,247)**	5.2 (3945)
Can go for a walk or some other exercise	30.4 (23,296)	**62.0 (47,446)**	7.6 (5820)
Can go out to work if you cannot work from home	21.1 (16,185)	**73.3 (56,085)**	5.6 (4292)
Can go out to help or provide care for a vulnerable person	20.5 (15,721)	**71.8 (54,976)**	7.7 (5864)
Can go out if you’re wearing a face covering	20.7 (15,885)	**72.7 (55,643)**	6.6 (5034)
Can go out to meet up with friends and/or family that you don’t live with, indoors *	12.4 (9495)	**83.0 (63,528)**	4.6 (3539)
Can go out to meet up with friends and/or family that you don’t live with, outdoors	16.3 (12,454)	**78.6 (60,147)**	5.2 (3960)
Can go out to spend time outdoors for recreational purposes (including to sit in parks, etc.)	24.7 (18,890)	**68.6 (52,525)**	6.7 (5147)
Can have someone who you don’t live with over to your home *	12.1 (9241)	**82.6 (63,229)**	5.3 (4092)
Can go out to get a test to see if you have coronavirus	**66.9 (51,229)**	23.3 (17,822)	9.8 (7510)
Should get a test, but can go out as normal while you are waiting for the result [included in waves 32 to 50, total n = 36,509]	13.8 (5051)	**79.1 (28,883)**	7.1 (2574)
Should take a rapid ‘lateral flow’ coronavirus test (results within 30 min) [included in waves 51 to 52, total n = 4043]	69.8 (2823)	**14.9 (604)**	15.2 (616)
Should take a lab-processed ‘PCR’ coronavirus test (results typically within a day or two) [included in waves 51 to 52, total n = 4043]	**77.0 (3114)**	9.4 (381)	13.6 (549)
Should take a test [included in waves 53 to 68, total n = 36,010]	**89.7 (32,303)**	6.4 (2318)	3.9 (1389)

* Included in derived key knowledge measure. Where percentages do not add to 100%, this is due to rounding errors. Where n does not add to the total n, this is due to the use of weighted data.

**Table 2 ijerph-20-01952-t002:** Factors associated with correctly identifying the key knowledge measure. Frequencies reported are based on weighted data. Adjusted odds ratios from the generalized estimating equation (GEE) are based on unweighted data.

Factor	Level	Key Knowledge Incorrect (Total n = 28,133), (%) n	Key Knowledge Correct (Total n = 47,345), (%) n	Adjusted Odds Ratio (95% CI) for Getting Key Knowledge Measure Correct ^†^	*p*
Survey wave	9 to 11 November 2020 (wave 32)	39.6 (801)	60.4 (1221)	Reference	-
	16 to 18 November 2020 (wave 33)	42.4 (868)	57.6 (1179)	0.93 (0.82 to 1.06)	0.28
	23 to 25 November 2020 (wave 34)	40.6 (815)	59.4 (1194)	0.98 (0.86 to 1.11)	0.73
	30 November to 2 December 2020 (wave 35)	40.8 (834)	59.2 (1208)	0.99 (0.87 to 1.12)	0.83
	7 to 9 December 2020 (wave 36)	46.2 (968)	53.8 (1126)	0.73 (0.65 to 0.83)	<0.001 *
	14 to 16 December 2020 (wave 37)	47.0 (948)	53.0 (1068)	0.74 (0.65 to 0.84)	<0.001 *
	21 to 23 December 2020 (wave 38)	48.0 (966)	52.0 (1046)	0.72 (0.63 to 0.81)	<0.001 *
	28 to 30 December 2020 (wave 39)	46.5 (909)	53.5 (1047)	0.75 (0.66 to 0.85)	<0.001 *
	4 to 6 January 2021 (wave 40)	42.6 (857)	57.4 (1153)	0.85 (0.75 to 0.96)	0.01
	11 to 13 January 2021 (wave 41)	39.9 (797)	60.1 (1199)	1.02 (0.90 to 1.16)	0.73
	25 to 27 January 2021 (wave 42)	41.5 (831)	58.5 (1170)	0.93 (0.82 to 1.06)	0.27
	8 to 9 February 2021 (wave 43)	39.5 (793)	60.5 (1215)	1.05 (0.93 to 1.19)	0.45
	22 to 23 February 2021 (wave 44)	38.0 (764)	62.0 (1245)	1.10 (0.97 to 1.25)	0.12
	8 to 9 March 2021 (wave 45)	40.6 (813)	59.4 (1191)	1.04 (0.92 to 1.18)	0.51
	22 to 23 March 2021 (wave 46)	41.4 (847)	58.6 (1200)	1.00 (0.88 to 1.13)	0.99
	6 to 7 April 2021 (wave 47)	39.2 (392)	60.8 (607)	1.08 (0.92 to 1.28)	0.33
	19 to 21 April 2021 (wave 48)	39.2 (389)	60.8 (604)	1.11 (0.95 to 1.30)	0.19
	4 to 5 May 2021 (wave 49)	38.6 (775)	61.4 (1232)	1.15 (1.01 to 1.31)	0.03
	17 to 19 May 2021 (wave 50)	40.3 (815)	59.7 (1205)	1.07 (0.95 to 1.21)	0.28
	1 to 2 June 2021 (wave 51)	40.5 (819)	59.5 (1203)	1.04 (0.92 to 1.17)	0.52
	14 to 16 June 2021 (wave 52)	41.0 (822)	59.0 (1185)	0.99 (0.88 to 1.12)	0.89
	28 to 29 June 2021 (wave 53)	39.5 (788)	60.5 (1206)	1.03 (0.91 to 1.17)	0.62
	26 to 27 July 2021 (wave 54)	40.4 (802)	59.6 (1185)	1.06 (0.94 to 1.20)	0.36
	9 to 10 August 2021 (wave 55)	39.1 (785)	60.9 (1221)	1.07 (0.95 to 1.21)	0.26
	23 to 24 August 2021 (wave 56)	40.9 (820)	59.1 (1186)	1.01 (0.90 to 1.15)	0.84
	6 to 7 September 2021 (wave 57)	41.2 (836)	58.8 (1193)	1.05 (0.93 to 1.19)	0.44
	20 to 22 September 2021 (wave 58)	24.7 (465)	75.3 (1421)	2.04 (1.79 to 2.32)	<0.001 *
	4 to 6 October 2021 (wave 59)	28.3 (534)	71.7 (1350)	1.80 (1.58 to 2.06)	<0.001 *
	18 to 20 October 2021 (wave 60)	25.8 (476)	74.2 (1372)	1.98 (1.73 to 2.26)	<0.001 *
	1 to 4 November 2021 (wave 61)	23.4 (506)	76.6 (1661)	2.14 (1.89 to 2.43)	<0.001 *
	15 to 17 November 2021 (wave 62)	31.8 (689)	68.2 (1477)	1.54 (1.36 to 1.74)	<0.001 *
	29 November to 1 December 2021 (wave 63)	32.0 (657)	68.0 (1397)	1.62 (1.43 to 1.84)	<0.001 *
	6 to 8 December 2021 (wave 63.5)	30.6 (625)	69.4 (1415)	1.63 (1.44 to 1.85)	<0.001 *
	13 to 16 December 2021 (wave 64)	31.3 (681)	68.7 (1493)	1.46 (1.29 to 1.65)	<0.001 *
	4 to 6 January 2022 (wave 65)	28.9 (643)	71.1 (1583)	1.71 (1.51 to 1.94)	<0.001 *
	17 to 20 January 2022 (wave 66)	27.0 (608)	73.0 (1645)	1.93 (1.70 to 2.19)	<0.001 *
	31 January to 2 February 2022 (wave 67)	30.8 (676)	69.2 (1516)	1.61 (1.42 to 1.82)	<0.001 *
	14 to 16 February 2022 (wave 68)	32.0 (719)	68.0 (1527)	1.48 (1.31 to 1.67)	<0.001 *
Age (per decade)	Range 16 to over 100 years	M = 43.5, SD = 18.9	M = 50.2, SD = 17.8	1.23 (1.22 to 1.25)	<0.001 *
Gender	Male	45.0 (16,474)	55.0 (20,167)	Reference	-
	Female	30.0 (11,556)	70.0 (26,995)	1.94 (1.88 to 2.01)	<0.001 *
	Prefer to self-describe	33.6 (73)	66.4 (144)	2.31 (1.67 to 3.19)	<0.001 *
	Prefer not to say	43.7 (31)	56.3 (40)	1.62 (0.97 to 2.68)	0.06
Nation	England	38.7 (24,562)	61.3 (38,860)	Reference	-
	Scotland	28.1 (1750)	71.9 (4468)	1.55 (1.45 to 1.66)	<0.001 *
	Wales	32.6 (1205)	67.4 (2492)	1.16 (1.07 to 1.26)	<0.001 *
	Northern Ireland	28.8 (617)	71.2 (1525)	1.55 (1.36 to 1.78)	<0.001 *
Index of multiple deprivation	1st quartile (least deprived) to 4th quartile (most deprived)	M = 2.6, SD = 1.1	M = 2.5, SD = 1.1	0.97 (0.96 to 0.99)	<0.001 *
Information from the Government about COVID-19 can be trusted	Strongly agree	44.4 (3708)	55.6 (4642)	Reference	-
	Agree	35.0 (8990)	65.0 (16,691)	1.39 (1.32 to 1.47)	<0.001 *
	Neither agree nor disagree	38.6 (7591)	61.4 (12,092)	1.33 (1.26 to 1.41)	<0.001 *
	Disagree	32.8 (4149)	67.2 (8489)	1.68 (1.58 to 1.79)	<0.001 *
	Strongly Disagree	40.4 (2946)	59.6 (4347)	1.33 (1.24 to 1.43)	<0.001 *
	Don’t know	46.1 (557)	53.9 (652)	1.11 (0.98 to 1.25)	0.12
To what extent do you agree or disagree that you have enough information from the Government and other public authorities with regards to…self-isolation	Strongly agree	34.6 (6885)	65.4 (13,020)	Reference	-
	Agree	34.5 (12,881)	65.5 (24,504)	1.00 (0.96 to 1.04)	0.88
	Neither agree nor disagree	50.2 (4995)	49.8 (4952)	0.63 (0.59 to 0.66)	<0.001 *
	Disagree	36.7 (2173)	63.3 (3747)	0.94 (0.88 to 1.01)	0.08
	Strongly Disagree	47.9 (820)	52.1 (893)	0.69 (0.62 to 0.77)	<0.001 *
	Don’t know	62.3 (379)	37.7 (229)	0.45 (0.38 to 0.53)	<0.001 *

† Adjusting for survey wave, age, gender, nation, index of multiple deprivation, trust in information about COVID-19 from the government, thinking you have enough information about self-isolation. * *p* < 0.05.

**Table 3 ijerph-20-01952-t003:** Endorsement of individual statements for knowledge of self-isolation for original and alternate wording of the item.

The Government Has Issued Advice on How People Should Help Prevent the Spread of Coronavirus if They Have Symptoms	If You Have Symptoms of Coronavirus, You [Total n = 1988]			If Someone Develops Symptoms of Coronavirus, They [Total n = 2011]			*p*
	True, % (n)	False, % (n)	Don’t Know, % (n)	True, % (n)	False, % (n)	Don’t Know, % (n)	
Should self-isolate	87.9 (1756)	7.1 (142)	5.0 (100)	87.1 (1751)	7.5 (150)	5.5 (110)	0.72
Should get a test, but can go out as normal while you/they are waiting for the result	15.3 (305)	77.5 (1550)	7.2 (144)	15.6 (314)	77.2 (1552)	7.2 (145)	0.95
Could be eligible for self-isolation payments (if you/they are on benefits and/or a lower income and cannot work from home)	61.3 (1225)	10.7 (214)	28.0 (560)	63.1 (1268)	9.7 (194)	27.3 (548)	0.40
Can go out if your/their symptoms are mild	8.3 (166)	85.9 (1717)	5.8 (116)	7.5 (150)	85.8 (1725)	6.8 (136)	0.30
Can go to the shops for groceries/pharmacy †	25.5 (510)	69.4 (1387)	5.1 (102)	22.0 (442)	71.5 (1437)	6.6 (132)	0.008 *
Can go to the shops for things other than groceries/pharmacy †	19.1 (381)	75.9 (1516)	5.1 (101)	17.8 (358)	76.2 (1533)	6.0 (120)	0.30
Can go for a walk or some other exercise	29.7 (594)	62.9 (1256)	7.4 (148)	25.6 (515)	65.9 (1325)	8.5 (171)	0.01 *
Can go out to work if you/they cannot work from home	23.5 (469)	70.1 (1401)	6.4 (128)	20.4 (411)	73.2 (1473)	6.3 (127)	0.06
Can go out to help or provide care for a vulnerable person	23.8 (475)	69.5 (1390)	6.7 (134)	21.1 (425)	69.5 (1398)	9.3 (188)	0.003 *
Can go out if you’re/they are wearing a face covering	23.4 (468)	70.8 (1415)	5.8 (115)	19.9 (401)	73.5 (1479)	6.5 (131)	0.02 *
Can go out to meet up with friends and/or family that you/they don’t live with, indoors †	12.6 (252)	82.7 (1654)	4.7 (93)	12.0 (242)	83.0 (1670)	5.0 (100)	0.78
Can go out to meet up with friends and/or family that you/they don’t live with, outdoors	22.0 (440)	73.6 (1471)	4.4 (88)	19.4 (391)	74.9 (1507)	5.6 (113)	0.04 *
Can go out to spend time outdoors for recreational purposes (including to sit in parks, etc.)	26.8 (535)	68.1 (1361)	5.1 (102)	24.3 (489)	69.6 (1400)	6.1 (123)	0.10
Can go out to get a test to see if you/they have coronavirus	67.5 (1350)	23.2 (463)	9.3 (186)	66.9 (1347)	22.8 (458)	10.3 (207)	0.57
Can have someone who you/they don’t live with over to your/their home †	11.6 (231)	83.5 (1670)	4.9 (98)	11.1 (223)	82.7 (1663)	6.2 (125)	0.18

† Included in derived key knowledge measure. * *p* < 0.05.

## Data Availability

Data are held by the Department of Health and Social Care and so no additional data are available.

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
