# Peer review of "Knowledge of Self-Isolation Rules in the UK for Those Who Have Symptoms of COVID-19: A Repeated Cross-Sectional Survey Study"

_ijerph, 2023, doi:10.3390/ijerph20031952_

Round 1

Reviewer 1 Report

I feel this is a very thorough review on topic of interest in the field of public health and health communication.

There are a few points to raise. 

1. While this is looks at knowledge at different time points of COVID self isolation it doesnt try to explore thoughts about COVID itself. For instance do they perceive Coronavirus a public health threat in general or a risk to them as an individual or their families as if they dont then self isolation would be seen as excessive and purposeless.

Following this it would make some sense that following 2 years of being told about isolation was there compliance fatigue with following self isolation measures. It would also be useful to consider how vaccination may have attenuated peoples understanding for the need to isolate. 

2. Your introduction is very good. Some minor points 'Since 12 March 2020)' please remove the bracket and ensure we say 12th March. 

I feel given the changes it would be easier to use a flow diagram to depict the evolving isolation rules during the time period you have selected. 

Please alter where you write (e.g. 18-20) .

3. Misinformation surrounding COVID has been rife. While you have assessed trust in said government and messages it would be useful to consider what health messages they perceived as being more trustworthy etc. 

4. The government payments is an interesting area to have acknowledged and read. I wonder how many used other governmental schemes whether that be for personal business or chose not to due to the effects of reduce income for their families.  

Author Response

We thank the reviewer for taking the time to read our manuscript and for their comments. We have responded to each point separately, below.

Point 1.

The reviewer is correct that beliefs and perceptions about COVID-19 may affect intention and adherence to self-isolation. This study focused on knowledge about self-isolation and factors associated with knowledge about self-isolation. As such, investigating the influence of these factors on knowledge about self-isolation was outside of the scope of this study.

This manuscript is part of a larger study that was carried out over the course of the pandemic (January 2020 to April 2022). As the reviewer suggests, levels of adherence to self-isolation may have increased and decreased over the course of the pandemic, due to many reasons including decreased motivation to isolate and decreased perception of necessity following the introduction of vaccination. We have published other manuscripts investigating adherence to isolation and factors associated with isolation (https://doi.org/10.1136/bmj.n608, https://doi.org/10.1111/bjhp.12576). In this study, we focused only on knowledge about self-isolation requirements and factors associated with knowledge.

Point 2.

We have corrected the minor errors you have identified:

  • Removing the bracket in the first paragraph of the introduction (now reads “Since March 2020,”)
  • We have added Figure 1, which is a flowchart showing the changes to self-isolation rules over the course of the pandemic.
  • Including author names and years for more clarity (now reads “Senedd Research 2022,19 Scottish Parliament 2022,20 Institute for Government 202221).

Point 3.

The reviewer is correct that there has been much misinformation about COVID-19. It would have been interesting to investigate credibility of individual messages. However, due to space limitations in the questionnaire, we were unable to do this. We have added this as a limitation to the study.

Point 4.

Investigating the rates of uptake of other governmental schemes is beyond the scope of this manuscript. The reviewer may find the following interesting:

  • https://www.gov.uk/government/collections/hm-treasury-coronavirus-covid-19-business-loan-scheme-statistics
  • https://www.gov.uk/government/collections/hmrc-coronavirus-covid-19-statistics
  • https://www.imf.org/en/Topics/imf-and-covid19/Policy-Responses-to-COVID-19

Reviewer 2 Report

The problem statement defined in the introduction is relevant and sufficiently specific. The presentation of the methodology and the measuring instrument is concise and clear, the latter is included by the authors. The presentation of the composition of the sample is informative. Due to the large sample size, the results are suitable for presenting differences according to the examined variables. The results of the study can also be used in practice: in case of a possible new wave of epidemics or other emergency situations, those social groups that require more information or a higher level of control can be identified.

Author Response

We thank the reviewer for taking the time to read our manuscript and for their kind comments.